

# Computer vision syndrome and associated factors among urban and rural bankers in Trinidad and Tobago

Kingsley Ekemiri[1], Devonte McKnight[1], Chioma Ekemiri[2],
Ngozika Ezinne[1], Henrietta Ashang[3], Virginia Victor[4],
Osaze Okonedo[5], Ayishetu Oshoke Shuaibu[6] and
Robin Seemongal-Dass[7]

[1] Optometry Unit, Department of Clinical Surgical Sciences, Faculty of Medical Sciences, The University of the West Indies, Tunapuna, Trinidad, Trinidad and Tobago
[2] Department of Health Promotion, Faculty of Education and Humanities, University of the West Indies, Health Promotion, Tunapuna, Trinidad, Trinidad and Tobago
[3] College of Health Technology, Dispensing Opticianry, Calabar, Cross River, Nigeria
[4] School of Nursing, Faculty of Medical Sciences, Mental Health Unit, Tunapuna, Trinidad, Trinidad and Tobago
[5] Courts Optical Optometry, Optometry Department, Tunapuna, Trinidad, Trinidad and Tobago
[6] Department of Optometry, University of Benin, Benin, Edo State, Nigeria
[7] Department of Clinical Surgical Sciences, University of the West Indies St. Augustine, St Augustine Campus, Trinidad and Tobago

Corresponding author
Kingsley Ekemiri,
iamekemiri@gmail.com

## ABSTRACT

**Background:** Modern workplace requirements in the banking sector require bankers to work on screens for more than 6 h a day, putting much stress and strain on their eyes, which leads to computer vision syndrome. Therefore, this study aimed to assess the prevalence of computer vision syndrome and associated factors among urban and rural bankers in Trinidad and Tobago.

**Methods and materials:** A cross-sectional design was applied to collect data from 399 bankers between April and June 2023. The collected data was entered into Excel worksheets and later uploaded to SPSS for further analysis. A variable with a *P*-value of 0.25 in binary logistic regression is a candidate for multi-variable logistic regression analysis. Finally, a variable with a *P*-value of 0.05 was used to declare statistical significance.

**Results:** A total of 371 participants were enrolled in this study, leading to a response rate of 92.9%. Of the total participants, 277 (74.7%) had computer vision syndrome. Working in rural areas (AOR = 2.69; 95% CI [1.41–5.13]) and using eyesight glasses (AOR = 0.57; 95% CI [0.33–0.97]) was associated with computer vision syndrome.

**Conclusion:** Despite being easily preventable, computer vision syndrome is substantially prevalent among bankers in Trinidad. The use of eye-sight glasses and the working area are significantly associated with computer vision syndrome. Therefore, it is necessary to improve workplace practices by encouraging the use of anti-glare screens and glasses for employees who work long hours on the computer.

## BACKGROUND

With the transformation to digitization, computers have been widely adopted as a useful tool that is efficient, reliable, and fast and it is now considered a necessity for all academic organizations and workplaces. From educational institutions to government and corporate offices to home offices, computer usage has exponentially increased (*Mi, 2016*). However, the shift to computers has resulted in a serious health concern known as the Computer Vision Syndrome (CVS). Firstly, CVS refers to a range of ocular symptoms caused by prolonged computer use, including eye strain, dry eyes, blurred vision, and headaches. These symptoms arise due to factors such as extended screen time, improper posture, poor workstation ergonomics, and inadequate lighting (*American Optometric Association, 1995*; *Ekemiri et al., 2022*).

Furthermore, computer use has been linked not only to ocular health problems such as CVS but also to broader health issues. Studies indicate that prolonged computer use can lead to vision-related disorders, including but not limited to eye strain and blurred vision. Additionally, it is associated with musculoskeletal issues like backaches due to prolonged sitting and poor posture, as well as tension headaches and general stress (*Sen & Richardson, 2007*).

CVS is negatively affecting the lifestyle and well-being of workers resulting in reduced quality of life, health issues, lower productivity and dissatisfaction with jobs (*Shantakumari et al., 2014*). Workplaces such as banks require workers to perform their daily tasks using computers and the long working hours put the bank workers at risk of developing CVS (*Boadi-Kusi et al., 2022*; *Kamal & Abd El-mageed, 2018*).

It is estimated that every year a minimum of one million new cases of CVS is reported and globally, approximately 60 million people are suffering from the digital eye strain which requires a serious evaluation and assessment of the underlying factors (*Ranasinghe et al., 2016*).

The increasing prevalence of CVS has become a worldwide concern for policymakers and researchers; which emphasizes the need of discussion on the importance of managing the underlying factors to keep up with the usage of computers (*Altalhi et al., 2020*).

Studies have shown that using of computers for 3 h per day increases the risk of development of CVS as compared to people who use computers for less than 3 h per day (*Klamm & Tarnow, 2015*). Furthermore, computer users who apply anti-glare filters and eyeglasses coatings over their screens complain less about ocular problems like CVS (*Lemma, Beyene & Tiruneh, 2020*; *Zayed et al., 2021*). CVS is a serious occupational hazard that needs to be addressed to ensure that the usage of computers in the digital era do not diminish the quality of life of users. So, this research aimed to assess the prevalence of CVSe and associated factors among urban and rural bankers in Trinidad and Tobago.

### Objectives of the study

✓ To determine the prevalence of CVS among urban and rural bankers in Trinidad and Tobago.

✓ To identify factors associated with CVS among urban and rural bankers in Trinidad and Tobago.

## METHODS AND MATERIALS

### Study design, setting, and period

The study used a cross-sectional study design and was conducted in Trinidad, one of the twin island of Trinidad and Tobago. The country boasts a population of approximately 1.4 million as of 2019, encompassing various ethnic backgrounds (*Kelly, 2023*; *Mohammed et al., 2021*). The study was conducted between April and July 2023.

### Source population

The target population for the study comprised bankers working across Trinidad in both rural and urban regions.

### Study population

The study participants were selected bankers working across Trinidad in rural and urban regions.

### Inclusion and exclusion criteria

The inclusion criteria for the selected staff members were as follows: having at least 1 year of working experience, exposure to excessive screening for over 6 months, and availability during the data collection period. Bankers with a history of acute or chronic eye diseases were excluded from this study.

### Sample size calculation and sampling procedure

The sample size was determined using the single population proportion formula. The following assumptions were applied: p is the prevalence of CVS (78.8%) (*Tesfaye et al., 2022*); d is the expected margin of error (4%), Z is the standard score corresponding to a 95% confidence interval, and, the risk of rejecting the null hypothesis (0.05). The final sample size was 399. Participants were selected using random sampling.

### Data collection tool and procedure

Self-administered structured questionnaires were used as the data collection tool. The questionnaire was carefully designed on the basis of the analysis of existing literature and consisted of five sections: social and demographic questions (six items), personal characteristics, behavioral factors, and screen use factors (14 items), knowledge assessment (28 items), ocular complaints (eight items), and resolution of complaints (four items). While self-administered questionnaires offer advantages like cost-effectiveness and convenience, potential issues such as respondent misinterpretation, response bias, and low response rates were addressed through pre-testing for clarity, ensuring anonymity for honesty, sending reminders for increased participation, and using a mix of question types

to capture comprehensive data. These measures are aimed at enhancing reliability and effectiveness of the data collection tool.

Branch managers were engaged to facilitate the data collection process, primarily by coordinating with participants to ensure timely completion of the questionnaires. They also played a crucial role in verifying participant eligibility and addressing any logistical issues that arose during data collection. Additionally, expert guidance was sought throughout the process to ensure the validity and reliability of the instrument.

## Data quality assurance

The questionnaire was initially developed in English, and its form and structure were reviewed by experts to ensure clarity and comprehensibility. Expert advice was incorporated to improve the questionnaire's quality and enhance respondent understanding, which increased the response rate.

## Operational definition

÷ Presence of glare on the computer screen: the existence of direct light sources on the computer screen due to unshaded windows with drapes/blinds (*Tesfaye et al., 2022*)

÷ Appropriate seating position: the face of the user is just at level with the computer screen (*Assefa et al., 2017*).

÷ Computer vision syndrome (CVS): the presence of at least one symptom (blurred vision, eye strain, eye fatigue, redness of the eyes, watery eyes, eye dryness, double vision, eye irritation, burning sensation, and headache) in one or two eyes at any time during the last 12 months (*Derbew et al., 2021*).

÷ Excessive screen use—Spending more than 2 consecutive hours in front of a screen without adequate breaks (*American Optometric Association, 2017*).

## Data analysis

The collected data were entered into Excel and then exported to SPSS version 26 for analysis. The data were cleaned and checked for any missing outliers and inconsistencies before analysis. Descriptive statistics were performed and are presented in the tables. Bivariate logistic regression analysis was performed to identify candidate variables for multivariate logistic regression at a $P$-value of 0.25 or less, and statistically significant associations were declared at $P$-value <0.05.

## Ethical consideration

The investigation into CVS among bankers of Trinidad, involving 371 participants obtained ethical clearance from the University of West Indies Ethical to carry out the study within its facilities with reference number CREC-SA.1175/09/2023. Subsequent to a thorough explanation of the study's procedures, informed consent was secured from the bankers. Adherence to the principles outlined in the Helsinki Declaration was rigorously maintained throughout the entirety of the study.

**Table 1 Sociodemographic characteristics of bankers, Trinidad.** This table provides a comprehensive overview of the sociodemographic attributes of bankers in Trinidad, capturing essential information related to their background. Key variables include age, gender, educational background, and other relevant factors that contribute to a nuanced understanding of the study participants. The data presented in this table forms the foundation for subsequent analyses, shedding light on the diverse socio-demographic composition of the banking professionals involved in the study.

| Variables | Category | Frequency | Percentage |
|---|---|---|---|
| Sex | Male | 136 | 36.7 |
| | Female | 235 | 63.3 |
| Age (years) | 18–23 | 42 | 11.3 |
| | 24–28 | 69 | 18.5 |
| | 29–33 | 63 | 16.9 |
| | 34–38 | 73 | 19.6 |
| | 39–43 | 41 | 11.0 |
| | >44 years | 83 | 22.3 |
| Ethnic background | African | 152 | 40.9 |
| | East India | 147 | 39.5 |
| | Mixed | 52 | 14.0 |
| | Other (Asian, Syrian, Caucasian & Hispanic) | 21 | 5.6 |
| Area type | Rural | 97 | 26.1 |
| | Urban | 274 | 73.9 |
| Education level | CSEC level | 66 | 17.8 |
| | CAPE level | 134 | 36.1 |
| | Bachelor's degree | 145 | 39.1 |
| | Master's degree and above | 26 | 7.0 |

# RESULTS

## Social and demographic characteristics of participants

A total of 371 participants were enrolled in this study, leading to a response rate of 92.9%. Most respondents were females 235 (63.3%), and 83 (22.3%) of the participants were aged greater than or equal to 44 years. The majority of the respondents were of African ethnicity 152 (40.9%). Moreover, the majority (274; 73.9%) and less than half (145; 39.1%) of the participants were urban dwellers and had an educational level of bachelor's degree (Table 1).

## Behavioral, personal, and usage characteristics of participants

Of the respondents, 282 (76.0%) reported having an appropriate seating habit and 304 (81.9%) used desktops for work. The screen level during use is reported to be above the eye level by 154 (41.5%) of the respondents. The majority of the respondents, worked for more than 6 h per day (273; 73.6%), took break for less than 20 min (285; 76.8%), and use eye glass while working (208; 56.1%). A total of 159 (42.9%) do not use an anti-reflective coating on their glasses. Furthermore, the light source in the office area for 313 (84.4%) patients was fluorescent. Nearly half of the respondents blink frequently while using a

**Table 2 Behavioral, personal, and usage characteristics of bankers, Trinidad.** This table presents a comprehensive overview of the behavioral, personal, and usage characteristics of bankers in Trinidad. It encompasses key factors influencing their work patterns, personal attributes, and utilization of digital technologies within the banking sector. The data provides valuable insights into the diverse aspects shaping the professional landscape of bankers in the Trinidadian context.

| Variables | Category | Frequency | Percentage |
|---|---|---|---|
| The type of seating position | Appropriate | 282 | 76.0 |
| | Not appropriate | 89 | 24.0 |
| Type of the used device | Laptop | 56 | 15.1 |
| | Desktops | 304 | 81.9 |
| | Cellular phone | 11 | 3.0 |
| Computer screen level | Above the eye level | 154 | 41.5 |
| | Below the eye level | 70 | 18.9 |
| | At the eye level | 147 | 39.6 |
| Hours of daily use | >6 h | 273 | 73.6 |
| | <6 h | 98 | 26.4 |
| Breaking habit | >20 min | 86 | 23.2 |
| | <20 min | 285 | 76.8 |
| Use of eyeglasses | Yes | 208 | 56.1 |
| | No | 163 | 43.9 |
| Type of coating | Anti-reflective coating | 156 | 42.0 |
| | Blue light blockers | 56 | 15.1 |
| | No coating | 159 | 42.9 |
| Blinks frequently | Yes | 183 | 49.3 |
| | No | 188 | 50.7 |
| Light source in the office | Natural light | 58 | 15.6 |
| | Fluorescent light | 313 | 84.4 |
| Brightness adjustment behavior | Yes | 150 | 40.4 |
| | No | 221 | 59.6 |
| Experience of glare | Yes | 103 | 27.8 |
| | No | 268 | 72.2 |
| Any systematic diseases | Yes | 41 | 11.1 |
| | No | 330 | 88.9 |
| Use of antiglare | Yes | 136 | 36.7 |
| | No | 235 | 63.3 |

computer. Brightness adjustment behavior was reported in 150 (40.4%) of the respondents, as 268 (72.2%) experienced no glare, 235 (63.3%) did not use ant-glare, and 330 (88.9%) had no confirmed systematic disease (Table 2).

## Knowledge of computer vision syndrome

The majority (247; 66.6%) of the respondents had poor knowledge of computer vision syndrome.

**Table 3 Occurrence of ocular complaints among bankers of Trinidad.** This table provides a comprehensive overview of the occurrence of ocular complaints among bankers in Trinidad. The data highlights the prevalence and distribution of various eye-related issues within the banking sector, offering valuable insights into the challenges faced by professionals in sustaining eye health in the workplace.

| Variables | Category | Frequency | Percentage |
|---|---|---|---|
| Eyestrain | Yes | 141 | 38.0 |
| | No | 230 | 62.0 |
| Blurred vision | Yes | 139 | 37.5 |
| | No | 232 | 62.5 |
| Eye redness | Yes | 89 | 24.0 |
| | No | 282 | 76.0 |
| Headache | Yes | 191 | 51.5 |
| | No | 180 | 48.5 |
| Dry eyes | Yes | 123 | 33.2 |
| | No | 248 | 66.8 |
| Eye fatigue | Yes | 202 | 54.4 |
| | No | 169 | 45.6 |
| Burning | Yes | 117 | 31.5 |
| | No | 254 | 68.5 |
| Irritation | Yes | 101 | 27.2 |
| | No | 270 | 72.8 |
| Double vision | Yes | 39 | 10.5 |
| | No | 332 | 89.5 |
| Watery eye | Yes | 78 | 21.0 |
| | No | 293 | 79.0 |
| Effort to relieve eye complaints | Yes | 218 | 58.8 |
| | No | 153 | 41.2 |

## Ocular complaints of the participants

More than half of 230 (62.0%) and 232 (62.5%) patients had no eyestrain or blurred vision. Two hundred and eighty-two (76.0%s) had never had eye redness, and 191 (51.5%) suffered from headaches. Dry eyes were experienced by one-third (123; 33.2%) of the respondents, eye fatigue by 202 (54.4%), burning sensation by 117 (31.5%), and irritation by 101 (277.2%) of the respondents. Moreover, more than three-fourth (332; 89.5%) of the bankers did not complain of double vision. More than half of the respondents (218) tried to resolve ocular problems (Table 3).

## Magnitude of computer vision syndrome

According to this study, approximately three-fourths of 277 (74.7%) bankers had developed CVS.

## Factors associated with the magnitude of computer vision syndrome

Nine variables in binary logistic regression with a *P*-value of 0.25 were candidates for multiple logistic regressions. Two variables (type of area and use of eyesight glasses) were

**Table 4 Factors associated with the magnitude of computer vision syndrome among Bankers of Trinidad (N = 371).** This table outlines the factors associated with the magnitude of Computer Vision Syndrome (CVS) among Bankers of Trinidad, based on a sample size of 371 individuals. Each factor is examined in relation to the severity of CVS experienced by the bankers, providing valuable insights into potential areas for intervention and mitigation strategies.

| Variables | Category | Magnitude of computer vision syndrome | | COR (95% CI) | AOR (95% CI) | P-value |
|---|---|---|---|---|---|---|
| | | Yes | No | | | |
| Type of seating position | Appropriate | 215 | 67 | 0.71 [0.42–1.24] | 0.77 [0.45–1.34] | 0.366 |
| | Not appropriate | 62 | 27 | 1 | 1 | 1 |
| Use of eyeglasses | Yes | 148 | 60 | 0.65 [0.40–1.05] | 0.57 [0.33–0.97] | 0.040*** |
| | No | 129 | 34 | 1 | 1 | 1 |
| Area type | Urban | 193 | 81 | 2.71 [1.43–5.13] | 2.69 [1.41–5.13] | 0.003*** |
| | Rural | 84 | 13 | 1 | 1 | 1 |
| Systemic disease | Yes | 34 | 7 | 5.75 [0.24–1.34] | 0.60 [0.25–1.46] | 0.268 |
| | No | 243 | 87 | 1 | 1 | 1 |
| Experience of glare | Yes | 68 | 31 | 1.15 [0.90–2.51] | 0.66 [0.39–1.12] | 0.127 |
| | No | 209 | 63 | 1 | 1 | 1 |
| Effort to relieve eye complaints | Yes | 168 | 50 | 1 | 1 | 1 |
| | No | 109 | 44 | 0.73 [0.46–1.18] | 1.57 [0.93–2.66] | 0.089 |
| Brightness adjustment behavior | Yes | 119 | 31 | 1 | 1 | 1 |
| | No | 158 | 63 | 0.65 [0.40–1.06] | 0.65 [0.39–1.08] | 0.102 |
| Use of antiglare | Yes | 96 | 40 | 1 | | |
| | No | 181 | 54 | 1.39 [0.86–2.25] | 0.81 [0.45–1.47] | 0.495 |
| Knowledge | Good | 48 | 94 | 1 | 1 | 1 |
| | Poor | 97 | 132 | 0.69 [0.41–1.16] | 1.50 [0.88–2.55] | 0.130 |

Note:
Values marked with *** indicate statistically significant differences between groups (P < 0.05).

associated with computer vision syndrome, with a *P*-value <0.05 (Table 4). Bankers who worked in rural areas were two times more likely to develop CVS (AOR = 2.69; 95% CI [1.41–5.13]) than those who worked in urban areas. Bankers who used eye-sighted glasses were 43% less likely to develop CVS (AOR = 0.57; 95% CI [0.33–0.97]) than those who did not use eye-sight glasses.

## DISCUSSION

There is no doubt that the advent of computer technology has led to the increased use of tablets, mobile phones, and computers in the workplace and in personal life. While these technologies have made life easier by making the world accessible, the excessive use of screens has been linked to several substantial health issues, especially eye problems. A CVS is a common and frequent diagnosis in people who use screens for a major part of their day (*Dessie et al., 2018*; *Chawla et al., 2019*; *Al Tawil et al., 2020*; *Turgut, 2018*).

In Trinidad, the banking sector employees do not give much preference to their health and safety, as the workplace environment has poor ergonomic and safety arrangements. The occurrence of strained eyes and CVS is, therefore, a frequent occurrence. In this study,

74.7% of bankers had CVS, which is similar to studies conducted in Ethiopia (74.6%, *Assefa et al., 2017*; 73%, *Derbew et al., 2021*), and a meta-analysis conducted in Ethiopia (73.7%) (*Adane, Alamneh & Desta, 2022*), and Ghana (71.2%) (*Boadi-Kusi et al., 2022*).

This finding is lower than studies conducted in Ethiopia, Gondar (90.20%; *Mersha et al., 2020*) and Egypt (85.2%; *Tesfaye et al., 2022*), this discrepancy might be due to variation in sample size, sociodemographic characteristics, and the knowledge level of this study participants is good.

In this study, the place of work was associated with CVS. Bankers who work in rural areas were two times more likely to develop computer vision syndrome (AOR = 2.69; 95% CI [1.41–5.13]) than those in urban areas. This might be because of health information and access to anti-glare coating, blue light shielding, and other eye strain-reducing mechanisms are relatively better in urban areas. Workers in rural areas have a greater workload than those in urban areas, which aggravates the issue (*Lemma, Beyene & Tiruneh, 2020*).

The use of eyeglasses is associated with CVS; in this study, bankers who used eyeglasses were 43% less likely to develop computer vision syndrome (AOR = 0.57; 95% CI [0.33–0.97]) than those who did not use eyeglasses. This result disagrees with that of the study conducted in Ethiopia. This might be because in technologically advanced cities, health-seeking and promotional behavior is better than in less advanced cities.

This study reveals significant disparities in the prevalence of CVS between rural and urban bankers, with rural workers being twice as likely to be affected. These differences are likely driven by variations in knowledge levels and health-seeking behaviors, as urban workers, with better access to health information and services, are more aware of preventive strategies such as regular eye check-ups and blue light filters. The findings underscore the need for targeted health interventions and public health education in rural areas, particularly in technologically intensive workplaces, to address these disparities and close the gap in occupational health research.

## CONCLUSION

In conclusion, this study highlights the significant prevalence of CVS among bankers in Trinidad, emphasizing its preventable nature. It establishes a strong link between eyesight glasses and workplace conditions, indicating the need for improved practices like anti-glare screens and proper use of glasses. The effectiveness of these glasses may depend on their quality and whether refractive errors are adequately corrected. Additionally, the research reveals rural-urban disparities in CVS prevalence, with rural workers being disproportionately affected, necessitating targeted interventions such as regular breaks and better workplace lighting. Ultimately, the study advocates for a comprehensive approach that combines individual responsibility with systemic changes to enhance the occupational health outcomes of banking professionals, particularly in rural areas.

### Limitations

First, the study was based on a large sample. It hence did not include any clinical examination, only self-reported data. Moreover, the use of a cross-sectional design makes

it difficult for the researcher to make a clear decision regarding the relation between CVS and various factors. Therefore, future researchers should use diverse inputs and conduct a longitudinal study involving some level of ophthalmic examination.

## ACKNOWLEDGEMENTS

We sincerely thank the banking sector in Trinidad and Tobago for their crucial support in this study. Your cooperation has been essential in assessing computer vision syndrome among bankers, and your commitment to providing a vision-safe environment is greatly appreciated. We value your efforts in improving workplace practices and look forward to ongoing collaboration.

### Funding

The authors received no funding for this work.

### Competing Interests

The authors declare that they have no competing interests.

### Author Contributions

- Kingsley Ekemiri conceived and designed the experiments, performed the experiments, prepared figures and/or tables, and approved the final draft.
- Devonte McKnight conceived and designed the experiments, performed the experiments, prepared figures and/or tables, and approved the final draft.
- Chioma Ekemiri conceived and designed the experiments, performed the experiments, analyzed the data, prepared figures and/or tables, authored or reviewed drafts of the article, and approved the final draft.
- Ngozika Ezinne analyzed the data, authored or reviewed drafts of the article, and approved the final draft.
- Henrietta Ashang conceived and designed the experiments, performed the experiments, prepared figures and/or tables, authored or reviewed drafts of the article, and approved the final draft.
- Virginia Victor conceived and designed the experiments, performed the experiments, prepared figures and/or tables, authored or reviewed drafts of the article, and approved the final draft.
- Osaze Okonedo conceived and designed the experiments, authored or reviewed drafts of the article, and approved the final draft.
- Ayishetu Oshoke Shuaibu analyzed the data, prepared figures and/or tables, authored or reviewed drafts of the article, and approved the final draft.
- Robin Seemongal-Dass analyzed the data, prepared figures and/or tables, authored or reviewed drafts of the article, and approved the final draft.

## Human Ethics

The following information was supplied relating to ethical approvals (*i.e.*, approving body and any reference numbers):

The University of West Indies Ethical approval to carry out the study within its facilities. (CREC-SA.1175/09/2023).

## Ethics

The following information was supplied relating to ethical approvals (*i.e.*, approving body and any reference numbers):

The University of the West Indies granted Ethical approval to carry out the study within its facilities (Ethical Application Ref: CREC-SA.1175/09/2023).

## Data Availability

The raw measurements are available in the Supplemental Files.

## Supplemental Information

Supplemental information for this article can be found online at http://dx.doi.org/10.7717/peerj.18584#supplemental-information.

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
