# Peer review of "Computer vision syndrome and associated factors among urban and rural bankers in Trinidad and Tobago"

_PeerJ, doi:10.7717/peerj.18584_

## Round 0.1 · original submission · Major Revisions

The manuscript has been reviewed by three experts in the field. Revisions are necessary before the manuscript is suitable for publication.

·

Basic reporting

The language was straightforward and precise, providing a clear and comprehensive background and introduction that established the context of the discussion.
The authors had the opportunity to strengthen the rationale for this study (line 96-99) considering its unique aspects. While similar studies have been conducted in Ethiopia, the inclusion of a rural component sets this study apart. This distinction could have been emphasized to provide a clearer justification for the research.

The literature cited in the study is well-referenced and relevant, notably the inclusion of recent research on CVS, particularly in the context of bankers, which is appropriate.

The article adheres to the fundamental Peer J standards, with the exception of the abstract's subheading style. It's worth noting that the main article begins with background information rather than an introduction. However, I speculate that the journal might be accommodating regarding this deviation.
The provided figures are relevant, of high quality, properly labeled, and contain detailed descriptions.
All raw data was supplied in compliance with the PeerJ policy.

Experimental design

This research is original and primary and falls within the journal's scope. Research questions are well-defined, relevant, and meaningful. The work clearly outlines how the research fills an identified knowledge gap. The two questions presented were well-defined, relevant, and meaningful. Additionally, the research successfully demonstrates how it fills the identified gap.
The research was conducted as proposed. The potential drawbacks of using a self-administered questionnaire should be acknowledged and addressed (lines 128-132).

While the description was detailed and allows for replication, it could be improved by providing a brief explanation of how the managers assisted (line 133) and where the expert advice was incorporated into the data collection instrument. This would make it clearer to participants how to respond to the question items. This would be relevant for the following:
Question 7 the operational definition would have been added in the bracket;
Questions 14 how would the participants know
Question 16 the options appear limited
Questions 24 to 38 required an explanation to ensure all understood
Question 39 appears subjective: how were participants able to answer objectively?
The earlier comment (about line 133) suggests that these observations might not be pertinent if managers had a guide to facilitate their assisting the participants effectively. If this was undertaken, it ought to be incorporated into the methodology. The discussion was skillfully integrated within the pertinent literature.

Validity of the findings

The study did not evaluate the impact or level of innovation and newness of its findings.

The study recommended explicitly articulating how additional clinical testing research can enhance the significance and validity of the study's conclusions

The provided underlying data is solid, statistically dependable, and has undergone rigorous quality control.

Conclusions are well-articulated and directly related to the research question. They are supported by the evidence gathered during the study and do not go beyond what the results indicate.

The conclusion strongly supported research question number 2, but its connection to research question number 1 was lacking.

Additional comments

The study presents valuable insights adding perspective to possible differences between urban and rural bankers with respect to CVS deserving of publication. Unfortunately, in some developing settings, CVS often does not receive the appropriate level of consideration.

·

Basic reporting

The manuscript “Computer Vision Syndrome and associated factors among Urban and rural bankers in the digitally driven modern banking landscape in Trinidad and Tobago” is good but needs a major review. The English language should be improved. Some suggestions are given below.
1. The introduction needs review specifically lines 75-80 need more clarification. Define computer vision syndrome concerning ocular symptoms. Separately clarify ocular health problems and general health issues related to the use of computers.
2. Line 85-93. Three consecutive single-sentence paragraphs. English language should be improved.
3. Line 79-98: I will suggest deleting “was” in the sentence “So this research was aimed to assess----”
4. Line 102-104: I will suggest deleting “in digitally driven modern banking landscape” in lines 102 and 104.
5. 4. Line 164-166. “Most respondents were females (235 (235 ( 235(63.3%), and 83(22.3%) of the participants were aged greater than or equal to 44 years”. This sentence needs clarification/correction for (235 (235 ( 235(63.3%), and 83(22.3%) ---
6. Line 167-168: In this paragraph, separate “--three-fourth (274, 73.9%) and less than half 145(39.1%) ------“concerning variables. In addition, 274 is inside parenthesis and 145 is outside parenthesis.
7. Line 170: In the sentence “More than three-fourths of 282 (282(76.0%), 304(81.9%)-----” 282 is repeated
8. Line 172-174: I will suggest dividing the sentence “The majority 273(73.6%), 285(76.8%), 208 (56.1%) work for more than six hours per day, take breaks for less than 20 “ minutes, and use eyeglasses while working, respectively in two sentences. Separate the majority 273(73.6%)--.
9. Line 178-179: “and 330 (88.9%) had no confirmed systematic disease (Table 2)” Systamic diseases are not mentioned in the methods section.
10. Line 181-182: “Two-thirds (247; 247(66.6%) of the respondents had poor knowledge of computer vision syndrome”. There is repetition of 247.
11. Line 186: Put percentage with (123).
12. Line 187: Correct “101 ( 101(277.2%)”
13. Line 188: Correct 332% in the sentence “Moreover, more than three-fourths (332%) of the bankers”
14. Line 196-197: “Two variables (type of area and use of Eyesight Glasses) were associated with computer vision syndrome”. This is not clear whether issues in the eyeglasses such as incorrect intra pupillary distance, decentration, prismatic effect in glasses, and quality of lenses were responsible or refractive error/presbyopia.
15. Line 200-201: “Bankers who used eye-sight glasses were 43% less likely to develop computer vision syndrome (AOR= 0.57; 95%CI 0.33- 0.97) than those who did not use eye-sight glasses”. The second part in this sentence” those who did not use eye-sight glasses” is not clear. Were these with refractive error but not using eye-sight glasses or these were with no refractive error?

16. Line 225: Replace the word damage by a suitable word in the sentence “especially eye damage”
17. Line 234-236: “Lower than studies--- needs grammar correction.
18. Table 4: “Knowledge” needs explanation. Knowledge about what?

Experimental design

Average

Validity of the findings

Average

Additional comments

As mentioned in earlier comments

Reviewer 3 ·

Basic reporting

The manuscript "Computer vision syndrome and associated factors among urban and rural bankers in the digitally driven modern banking landscape in Trinidad and Tobago" aims to analyze the prevalence of CVS among bankers in Trinidad and Tobago.

This topic is novel and little is known about the prevalence of CVS in this population. The authors designed a questionnaire for collect the data and the sample size is big. In my opinion, the weak point of this study is the design of the questionnaire. Some of the questions are vague or subjective (e.g. the seating position is defined as ‘appropiate or not’, the source of lighting as “natural or fluorescent”; or the brightness adjustment). Moreover, as for electronic devices, this questionnaire focuses on computer use. However, most participants may use smartphones or tablets, which may increase the daily hours of use and therefore the visual symptomatology.

Given that the sample size is large and that the authors have collected a lot of information on this topic, a much more thorough analysis could improve this manuscript to make it scientifically interesting.

Experimental design

Taking into account the data that has been compiled with this questionnaire, I consider that the data analysis can be improved:

I would recommend the authors to describe deeply the inclusion and exclusion criteria. For instance: in line 118: Working experience (with computer?); What do the authors mean with “excessive screening”? Specify the daily hours of computer use.

Demographic data are shown descriptively; is there a relationship between demographic data and CVS-related symptoms? (age, gender, ethnic background, area, education level). Moreover, I am not sure how useful it is to include so many age ranges, maybe 3-4 ranges would be enough. This statistical analysis can be included in table 1.

I would recommend the authors to merge tables 2 and 4.

The questionnaire asks about the severity of symptoms, but the results only show whether or not participants suffered from each symptom. A more in-depth analysis must be included.

Finally, it would be very interesting to carry out a multivariate analysis of variables that may influence CVS. For instance, the relationship between visual symptoms and: Gender and age; type of computer and hours of use; lighting and glare; breaks and hours of use...

Validity of the findings

In general, It is recommended that the authors describe the results of the study more precisely and include the statistical analysis described above. Some examples that have been found across the results:

Of the 371 respondents, 152 were African, and the authors described that “most respondents were African (40.9%)”. Since this percentage does not reach more than 50%, I would not say “most”.

The authors used ‘three quarters’ in all results. I would recommend the authors to be more accurate and describe the exact percentage in each section.

Additional comments

Technical errors are found throughout the text. Some examples from the first part of the manuscript:

Line 46: The abbreviation "hours" (h) should be defined.
Line 47: Replace “le ads” by "leads"
Line 57: Start the sentence "Working in rural areas" with capital letter
Line 59: This sentence seems uncomplete: (.....) Was associated with a higher prevalence of CVS?
Line 78: The definition of CVS can be improved and more original references must be included. Moreovere, “Ekemiri et al., 2022 did not define CVS, this is a study about the influence of COVID-19 pandemic in CVS.
Line 81: Replace "CVS are..." by "CVS is..."
Line 99: End the sentence with a dot.
Line 102: Remove "/".

---

## Round 0.2 · Minor Revisions

It has been re-reviewed by three experts in the field. Final revisions are necessary before the manuscript is suitable for publication.

·

Basic reporting

The language remains straightforward and precise, making the paper accessible. The authors successfully strengthened the rationale for the study by incorporating the distinction between rural and urban settings, which adds a clear justification for the research. This adjustment provides the necessary context and makes the study's unique aspects more evident.

The authors have appropriately revised the abstract's subheading style, as per my initial suggestion. Additionally, beginning with background information instead of a traditional introduction appears to align with journal standards. The quality of figures, their labeling, and the descriptions remain adequate.

No further adjustments are needed here. The revisions are satisfactory.

Experimental design

I appreciate the improved discussion regarding the limitations of using a self-administered questionnaire. The adjustments made to this section, such as emphasizing the steps taken to ensure data reliability, are well executed. The explanation about pre-testing, anonymity, reminders, and mixed question types strengthens the methodology.

The authors have added some clarity on how the managers and expert advice contributed to the data collection process. However, I would suggest that further detail be provided to explain how this support was systematically incorporated. For instance, elaborating on any specific training or guides given to managers would enhance transparency.

Response to Specific Questions (7, 14, 16, 24-38, 39): The authors' responses to the feedback on individual questions were adequate. While I initially suggested including an operational definition for Question 7, the decision to maintain clarity by avoiding complexity is reasonable.
The verbal explanations provided for Questions 14 and 24-38, and the structured response scale used for Question 39, sufficiently address the concerns raised. The authors have justified the limited response options in Question 16 to align with the study’s scope, which is acceptable.

The improvements made are satisfactory, but additional clarification on the manager's role and expert guidance could still benefit the methodology section.

Validity of the findings

While the study demonstrates statistical rigor and reliability, the level of innovation in the findings is still somewhat understated. I recommend including a more explicit discussion of the study’s contribution to the field, especially in terms of its rural-urban comparison and its implications for clinical practice. This would strengthen the overall impact of the research.

Clinical Testing and Research Recommendations: The revised recommendations regarding further clinical testing are appropriate and add value to the study's conclusions.
The findings are valid and statistically robust, with only minor improvements needed to highlight the study's innovative contributions.

Additional comments

The authors have addressed my earlier concern about the connection between the conclusions and research question 1. Both the abstract and conclusion sections have been revised to reflect this, making the conclusions more balanced and directly related to both research questions.

·

Basic reporting

The manuscript “Computer Vision Syndrome and associated factors among Urban and rural bankers in the Digitally Driven Modern Banking Landscape in Trinidad and Tobago” is good; however, I will suggest some changes. Such as
1. All bankers use computers. The authors do not compare the CVS in bankers working in the digitally driven modern banking with non-digitally bank systems, therefore, I will suggest deleting “in the digitally driven modern banking landscape” from the title and objectives. The title “Computer Vision Syndrome and associated factors among urban and rural bankers in Trinidad and Tobago” will be short and comprehensive.
2. Writing needs editing. Some paragraphs are long and some are very short. For instance Line 249-250, a one-sentence paragraph.
3. References need to be corrected. For instance
i. Line 131- “The questionnaire was carefully designed based on the analysis of existing literature and---“. Need reference.
ii. References style is not consistent. For instance, in the first reference, the journal name is full (journal of Computer Science and Engineering). In contrast, in the third reference, the journal's name is abbreviated (J Hum Ergol).

Experimental design

This is a cross-sectional survey.

Validity of the findings

In conclusion, the authors say “The use of eye-sight glasses and the working area are significantly associated with this condition. Therefore, it is necessary to improve workplace practices by encouraging the use of anti-glare screens and glasses for employees who work long hours on the computer”. The authors did not mention here whether it was due to the quality/no anti-glare coating of glasses or due to the reason that refractive errors were not properly corrected

Additional comments

Generally, the manuscript is good.

Reviewer 3 ·

Basic reporting

The authors have correctly responded to the comments of all reviewers. In my opinion, the manuscript has improved and meets the journal's criteria for publication.

Experimental design

Methodology is well explained and clearer.

Validity of the findings

No comment.

Additional comments

Attached are some comments on technical mistakes that have been found throughout the text:

• Line 60: Delete “Computer vision syndrome”, it has been already abbreviated in line 59.
• Line 70: Replace “are” by “is (CVS is…)
• Lines 86, 89, 91, 202, 208, 210, 231, 244, 251, (…): Abbreviate CVS
• Lines 178, 181, 192: I would recommend the authors to delete “three-fourth”.
o Line 181: This sentence is difficult to read. I suggest: Of the respondents, 282 (76.0%) reported having an appropriate seating habit and 304 (81.9%) used desktops for work.
• Lines 183-185: This sentence does not make sense to me; I would recommend making the same change I suggest in line 181.
• Line 249: Merge this sentence with the paragraph above.

---

## Round 0.3 · Minor Revisions

I have now had the opportunity to read your revised manuscript, and your responses to the reviewers' comments. I believe that you have addressed the concerns raised, and I am happy in principle to accept your manuscript for publication in PeerJ. The decision has however been listed as "minor revision" because before scheduling it for publication, I would be grateful if you could address some minor points that are shown.

·

Basic reporting

The authors have made the requested revisions based on my review comments
Language and Clarity: The distinction between rural and urban settings is now clearer, providing stronger rationale and context for the study.
Abstract and Subheading Style: The abstract has been revised to include the necessary subheadings, starting with the background instead of a traditional introduction, which aligns with the journal's standards.

Experimental design

Limitations of Self-Administered Questionnaire: The limitations have been discussed in greater detail, highlighting measures taken to ensure data reliability, such as pre-testing, anonymity, and reminders, as requested.
Manager and Expert Support: Additional information was provided regarding the role of managers and experts in the data collection process. .

Validity of the findings

Statistical Rigor and Contributions: The findings are presented with statistical accuracy and rigor. While the rural-urban differences and their implications for clinical practice are noted, further emphasis on the study’s innovative contributions could be beneficial.

Additional comments

The conclusions now explicitly relate to both research questions, ensuring a more balanced and connected final section. This revision directly addresses your feedback.

Overall, the revisions made by the authors align well with my review comments, and the manuscript now meets the requirements specified.

The requested changes have been implemented adequately, and no further review appears necessary.

·

Basic reporting

The authors have. The revisions are satisfactory.

Experimental design

The improvement made in the manuscript is satisfactory

Validity of the findings

Improved

Additional comments

Nill

Reviewer 3 ·

Basic reporting

The authors have correctly responded to the comments of all reviewers.

The manuscript has improved and meets the journal's criteria for publication. However, technical mistakes are still found across the text that should be revised carefully. Some examples:
Line 60: Remove ( ) from "(CVS)".
The references have various formats throughout the text, it is important to review them and put them all in the same format (Lines 240-242: [ ] format).
It would be also important to review the bibliography. For example, in line 317, the authors have not been included properly.

Experimental design

No comment

Validity of the findings

No comment

---

## Round 0.4 · accepted · Accept

I believe that you have addressed the concerns raised, and I am happy to accept your manuscript.